# Genome-wide identification, phylogenetic, and expression analysis under abiotic stress conditions of LIM gene family in *Medicago sativa L.*

Lili Nian[1], Xuelu Liu[1,2]*, Yingbo Yang[2]*, Xiaolin Zhu[3], Xianfeng Yi[4], Fasih Ullah Haider[2]

**1** College of Forestry, Gansu Agricultural University, Lanzhou, China, **2** College of Resources and Environmental Sciences, Gansu Agricultural University, Lanzhou, China, **3** College of Agronomy, Gansu Agricultural University, Lanzhou, China, **4** The Animal Husbandry Research Institute of Guangxi Zhuang Autonomous Region, Nanning, China

* liuxl@gsau.edu.cn (XL); yyb_17929@163.com (YY)

**Data Availability Statement:** All relevant data are within the paper and its S1–S3 Files and S1–S5 Tables.

## Abstract

The LIM (Lin-11, Isl-1 and Mec-3 domains) family is a key transcription factor widely distributed in animals and plants. The LIM proteins in plants are involved in the regulation of a variety of biological processes, including cytoskeletal organization, the development of secondary cell walls, and cell differentiation. It has been identified and analyzed in many species. However, the systematic identification and analysis of the LIM genes family have not yet been reported in alfalfa (*Medicago sativa* L.). Based on the genome-wide data of alfalfa, a total of 21 LIM genes were identified and named MsLIM01-MsLIM21. Comprehensive analysis of the chromosome location, physicochemical properties of the protein, evolutionary relationship, conserved motifs, and responses to abiotic stresses of the LIM gene family in alfalfa using bioinformatics methods. The results showed that these MsLIM genes were distributed unequally on 21 of the 32 chromosomes in alfalfa. Gene duplication analysis showed that segmental duplications were the major contributors to the expansion of the alfalfa LIM family. Based on phylogenetic analyses, the LIM gene family of alfalfa can be divided into four subfamilies: αLIM subfamily, βLIM subfamily, γLIM subfamily, and δLIM subfamily, and approximately all the LIM genes within the same subfamily shared similar gene structure. The 21 MsLIM genes of alfalfa contain 10 Motifs, of which Motif1 and Motif3 are the conserved motifs shared by these genes. Furthermore, the analysis of cis-regulatory elements indicated that regulatory elements related to transcription, cell cycle, development, hormone, and stress response are abundant in the promoter sequence of MsLIM genes. Real-time quantitative PCR demonstrated that MsLIM gene expression is induced by low temperature and salt. The present study serves as a basic foundation for future functional studies on the alfalfa LIM family.

**Funding:** This research was financially supported by the National Natural Science Foundation of China (31601984, awarded to YY); the Scientific Research Start-up Funds for Openly-Recruited Doctors of Gansu Agriculture University (2017RCZX-32, awarded to YY); The Youth Science and Technology Fund of Gansu Province (20JR5RA014, awarded to YY); and Basic Scientific Research Funds of Guangxi Institute of Animal Sciences (2017-20, 2018-27, awarded to XY).

**Competing interests:** The authors have declared that no competing interests exist.

# 1 Introduction

Transcription factors, also known as trans-acting elements, regulate the expression of stress-inducible genes by directly or indirectly binding specifically the cis-elements in promoter regions. The family of LIM proteins possesses a cysteine-rich zinc-binding domain named as LIM domain [1]. The name of the LIM domain is derived from the acronyms of LIN11, ISL1, and MEC3, which are identified from animals containing LIM domain proteins [2–4]. The LIM domain is widely distributed and highly conserved in eukaryotes. The universal make-up of the LIM domain comprises two zinc fingers linked together with a short two-amino acid spacer [5]. In plants, HaP-LIM1 was first isolated from sunflower (*Helianthus annuus* L.) pollen [6], and then in tobacco (*Nicotiana tabacum* L.) [7], lily (*Lilium longiflorum Thunb* L.) [8], Arabidopsis (*Arabidopsis thaliana* L.) [9], cotton (*Gossypium hirsutum* L.) [10], tomato (*Solanum lycopersicum* L.) [11, 12], and foxtail millet (*Setaria italica* L.) [13], and other species have successively discovered LIM protein.

The LIM in plants comprises two sub-families based on the number of LIM domains: single LIM sub-family (DA1/DAR) and double LIM sub-family (2LIMs). The 2LIM protein contains two LIM domains separated by 40–50 amino acid residues. Structurally, the 2LIM subfamily in the plant is similar to the domains of cysteine-rich protein (CRP) family members in animals, but there are some differences. Such as the plant 2LIM protein contains a variable short C-terminus and lacks a Glycine region. Based on their expression mode, 2LIM proteins are divided into two types: WLIM (WLIM1 and WLIM2), which is widely expressed in various tissues, and PLIM (PLIM1 and PLIM2), which is widely expressed in pollen tubes [14]. The DA1 & DAR sub-family is present only in plants and contains a conserved LIM domain [15]. According to the amino acid sequence, DA1 & DAR proteins are divided into two types: Class I and Class II. So far, DA1 & DAR LIM proteins have been identified only in land plants. The subcellular localization of plant LIM protein is the same as that of animals, with three simultaneous localizations of cytoplasm, nucleus, and nucleoplasm. In animals, the LIM protein located in the nucleus is mainly involved in the transcriptional regulation of developmental genes [16], while the LIM protein located in the cytoplasm is mainly involved in the regulation of actin cytoskeleton formation as actin-binding protein (ABP) [17]. Tobacco NtWLIM1, sunflower WLIM1, and Arabidopsis LIM protein families all show the characteristics of simultaneous localization in the nucleus and cytoplasm [18]. Tobacco NtWLIM1 protein can specifically bind to PAL-box to activate key enzymes in the phenylpropane metabolic pathway and participate in the regulation of lignin biosynthesis [19]. NtWLIM2 participates in the regulation of cell proliferation and cell cycle progression by activating the expression of histone genes [20]. In addition, it was found that the DA1 & DAR subfamily determines organ size and plant autoimmune response [21, 22]. In recent years, studies have shown that up-regulation of LIM gene expression can respond to abiotic stresses, including drought, salinity, and hormones, indicating that LIM genes may play a key regulatory role in the resistance of plants to various stress responses [15, 23, 24].

As a perennial legume forage of the genus Medicago, alfalfa has the characteristics of high yield, good palatability, and strong adaptability, and has a long cultivation history, and is widely planted [25]. Alfalfa can not only be used as feed, but also has the functions of maintaining water and soil, improving soil, and protecting the ecological environment. Therefore, cultivating tolerant alfalfa varieties is an economic and effective way to resist adversity environment. This experiment uses bioinformatics analysis to identify the alfalfa LIM gene family at the genome-wide level, and further analyzes the gene structure, chromosome distribution, promoter cis-acting elements. Using qRT-PCR analyze the expression of LIM in the leaves of alfalfa at different treatment time points under salt and low-temperature stress. This study can provide a theoretical basis for future research on the function of the alfalfa LIM gene.

## 2 Materials and methods

### 2.1 Identification and data collection of alfalfa LIM transcription factor family genes

The genome-wide data of alfalfa is obtained from the "Alfalfa Breeder's Toolbox" (https://www.alfalfatoolbox.org/). In order to identify all members of the LIM family genes in alfalfa, we downloaded the Arabidopsis LIM gene family sequence and annotation information from the TAIR (http://www.arabidopsis.org/) database. Using the TB tools software [26], based on the reference sequence of the Arabidopsis LIM gene family, the possible LIM family sequence in alfalfa was retrieved through Pfam (http://pfam.xfam.org/family), NCBI-CDD (https://www.ncbi.nlm.nih.gov/cdd/) and SMART (http://smart.embl-heidelberg.de/) online tools to predict protein conserved domains.

### 2.2 Basic physical and chemical properties and evolutionary analysis of LIM protein

The online software ExPASy (https://web.expasy.org/protparam/) was used to analyze the basic physical and chemical properties of the protein encoded by the alfalfa LIM gene, including the number of amino acids, molecular weight, theoretical isoelectric point (pI), and the average value of hydrophilicity (GRAVY), subcellular location is predicted by Psort-Prediction (http://psort1.hgc.jp/form.html) and Cell-PLoc (http://www.csbio.sjtu.edu.cn/bioinf/Cell-PLoc-2/).

We used 10 species to study the evolutionary relationship between alfalfa LIM genes and other plant LIM genes, including Arabidopsis (*Arabidopsis thaliana* L.), tobacco (*Nicotiana tabacum* L.), barrel medic (*Medicago truncatula* L.), soybean (*Glycine max* L.), tomato (*Solanum lycopersicum* L.), quinoa (*Chenopodium quinoa* L.), maize (*Zea mays* L.), rice (*Oryza sativa* L.), wheat (*Triticum aestivum* L.), and identified alfalfa LIM genes. Fast, scalable generation of LIM protein multiple sequence alignments using Clustal Omega (https://www.ebi.ac.uk/Tools/msa/clustalo/). At the same time, MEGA 7 software was used to construct the phylogenetic tree with the Maximum Likelihood method (ML), and the bootstrap was repeatedly set to 1000. After the phylogenetic tree is constructed, the members of the family are classified according to the classification standards of Arabidopsis, tobacco, and rice.

### 2.3 Chromosome location and gene duplication analysis

Using the annotation information of the LIM gene in the alfalfa genome database, the distribution of the members of the alfalfa LIM family on the 32 alfalfa chromosomes was analyzed. On the plant genome duplication database server (http://chibba.agtec.uga.edu/duplication/index/locket), the duplicate gene pairs are detected. The amino acid sequence of the partially repeated MsLIM gene was predicted using Clustalw software. Synonymous substitution rate (Ks) and non-synonymous substitution rate (Ka) and Ka/Ks are calculated using online tools Clustal Omega and PAL2NAL (http://www.bork.embl.de/pal2nal/).

### 2.4 Gene structure and conservative motif analysis

According to the GFF annotation file information of the whole genome of alfalfa, use the online website Gene Structure Display Server (GSDS) (http://gsds.cbi.pku.edu.cn/) to obtain the visualized picture of exon-intron [27]. The conserved motifs of alfalfa LIM protein were analyzed using Multiple Expectation Maximization for Motif Elicitation (MEME Suite) (http://meme-suite.org/), and the number of Motifs was set to 10 [28].

## 2.5 Analysis of promoter cis-acting elements

The 2000bp sequence upstream of the LIM gene was used as the promoter of the alfalfa LIM gene. The Plant CARE (http://bioinformatics.psb.ugent.be/webtools/plantcare/html/) database was used to predict and organize promoter cis-acting elements, and display them in the form of graphs.

## 2.6 Construction of protein interaction network diagram and three-dimensional structure prediction

The model plant Arabidopsis thaliana was used as a background to predict the protein network structure interaction of alfalfa LIM.STRING (http://STRINGdb.org/) software is used to construct a protein network structure diagram (confidence limit is 0.4) [29]. At the same time, we used SWISS-MODEL [30] (https://swissmodel.expasy.org/interactive) provided by the online software ExPaSy to model the three-dimensional structural homology of the protein spatial model of the MsLIM gene family of alfalfa.

## 2.7 Planting and stress treatment of alfalfa material

Select alfalfa seeds with the same shape and full grains, cut the seed coat and allowed to germinate in an artificial climate chamber. The culture temperature is 24°C, the relative humidity is 80%, and the photoperiod is 16/8 h. The seedlings 5 days after germination were transferred to a plastic box and cultured with MS solid medium under the same conditions. Four-week-old alfalfa with consistent growth status was subjected to stress treatment, and 100mmol/L NaCl and 4°C were used for salt stress and low-temperature stress, respectively. The treatment time was 0 h, 3 h, 6 h, 9 h, and 12 h. The mature leaves were sampled quickly and put into liquid nitrogen, 3 copies for each treatment, and then placed in -80°C cryopreservation for subsequent quantitative experiments by following protocol mentioned by.

## 2.8 RNA extraction and qRT-PCR detection

Use Shenggong's UNIQ-10 column Trizol total RNA extraction kit to extract total RNA from each sample, and use Nano-Drop 2000 UV spectrophotometer to detect RNA quality and concentration. Use M-Mu LV first-strand cDNA synthesis kit reverse transcription RNA to obtain cDNA. After detecting the concentration, uniformly dilute to 100 ng/ul as the q RT-PCR reaction template. Specific primers were designed using Perl Primer v1.1.21 software [31]. The primers used in the experiment are shown in the S1 Table. Use Shenggong's 2x SG Fast q PCR Master Mix kit, the reaction system is 20 μl, the PCR reaction program is 95°C pre-denaturation for 10 min, and then 40 cycles including 95°C denaturations for 15 s and 60°C annealings for 1 min, the instrument used For Applied Biosystems 7500, the experimental results are processed by the $2^{-\triangle\triangle Ct}$ method [32]. Each experiment was repeated three times with independent RNA samples. Analysis of variance (ANOVA) of the relative expression level of each gene at different sampling points under each abiotic stress treatment was carried out following a generalized linear model using SPSS22 statistical software. Significant differences in mean values at different sampling times were determined by Tukey's pairwise comparison tests, as indicated by different letters in the figures. The graphical representation of the experimental findings was produced by using Graphpad.

## 3 Results

### 3.1 Basic physical and chemical properties

Using the Arabidopsis LIM protein sequence (data in S2 Table) and the alfalfa genome sequence to perform BLAST comparison to extract the CDS (data in S1 File) and protein

sequence (data in S2 File) of the alfalfa LIM gene, after de-redundancy and SMART identification [26], 21 alfalfa LIM genes were finally identified and named as MsLIM01-M-sLIM21(data in S2 Table). The size of the coding sequences (CDSs) of the members of the MsLIM gene family is between 433–1113 nucleotides. The amino acid lengths of the proteins encoded by these genes vary from 154aa to 212aa, the molecular weight ranges from 17.14kDa to 23.73 kDa, and the PI variation range of the encoded protein from 6.20–9.16, the GRAVY value of all MsLIM proteins is less than 0, indicating that these proteins are hydrophilic. Prediction of the subcellular location showed that most MsLIM genes were located in the cytoplasm (16 MsLIM), and a few were located in mitochondria and peroxisomes.

## 3.2 The construction of phylogenetic tree

To explore the evolutionary relationship between the LIM gene family and predict its classification in alfalfa, the LIM conserved protein sequences of 9 different species (Arabidopsis thaliana, tobacco, barrel medic, soybean, tomato, quinoa, corn, rice, and wheat (data in S3 Table) are used for phylogenetic tree construction to study the evolutionary relationship between this family and other species LIM. According to the evolutionary relationship of LIM and the classification of Arabidopsis and rice, alfalfa LIM can be divided into four subfamilies, namely αLIM, βLIM, γLIM, and δLIM (Fig 1). The subfamily δLIM includes 10 LIM proteins, which is the largest subfamily in LIM, while the numbers in the other three subfamilies are 4 (αLIM), 4 (βLIM), and 3 (γLIM). It can be seen from Fig 1 that the closest relationship with the alfalfa LIM gene family is the barrel medic and soybean, which belong to the legume family, indicating that alfalfa, barrel medic, and soybean are highly conserved during the evolution process, which can be passed through the studied species. The function

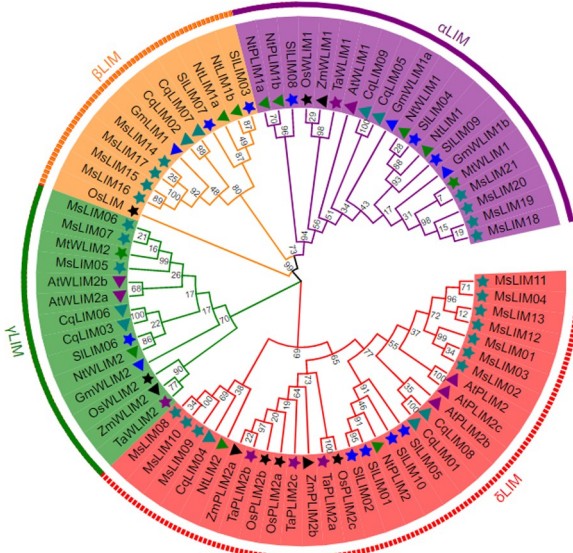

**Fig 1. Phylogenetic tree showing the relatedness of alfalfa LIM proteins to that of Arabidopsis, tobacco, barrel medic, soybean, tomato, quinoa, maize, rice, and wheat.** A species acronym was added before each LIM protein name: At, *Arabidopsis thaliana*; Nt, *Nicotiana tabacum*; Mt, *Medicago truncatula*; Gm, *Glycine max*; Sl, *Solanum lycopersicum*; Cq, *Chenopodium quinoa*; Zm, *Zea mays*; Os, *Oryza sativa*; Ta, *Triticum aestivum*. (The information of the LIM family members is listed in the supporting information (S2 Table, Among them, the LIM amino acid sequence of Arabidopsis is from TAIR (http://www.arabiodpsis.org). The LIM amino acid sequences of tobacco, barrel medic, soybean, and tomato are from NCBI (https://www.ncbi.nlm.nih.gov/). The LIM amino acid sequences of quinoa, corn, rice, and wheat are from Phytozome (https://phytozome.jgi.doe.gov/pz/portal.html)).

of the LIM gene family is used to speculate the function of alfalfa LIM. At the same time, it was found that the LIM genes of monocotyledonous plants clustered into small branches separately, indicating that there are certain differences in the evolutionary process of monocotyledonous and dicotyledonous LIM gene families. Different evolutionary rates lead to the diversity of LIM gene functions, and the subfamily formed later produces new subfamily-specific functions after differentiation. Therefore, the functions of LIM genes in different subfamilies are different.

### 3.3 Chromosome location and gene duplication analysis

According to the annotation information in the alfalfa genome (data in S3 File), we found that 21 alfalfa LIM genes are distributed on 17 of the 32 chromosomes of alfalfa. There are 4 LIM genes on each of chromosomes 1, 2, and 4, which are located on chromosomes 1.1, 3.1, 7.1, 8.1, 1.2, 3.2, 4.2, 8.2, 1.4, 4.4, 7.4, and 8.4 respectively, and on chromosomes of 3 groups. There are 5 LIM genes located on chromosomes 1.3, 3.3, 4.3, 7.3, and 8.3, and no distribution of LIM genes is found on other chromosomes. In addition, except for the four homologous chromosomes of chromosome 8 which contain 2 LIM genes, all the other chromosomes contain 1 LIM gene. Since alfalfa is a 4-ploid homologous, the distribution of some homologous genes on homologous chromosomes is almost the same. Five or fewer genes located within 100 kb of the same chromosome are usually considered as tandem repeat genes. So there are no tandem repeat gene pairs in this study. The online tools Clustal Omega and PAL2NAL were used to analyze the MsLIM gene replication events, and a total of 8 replicated gene pairs were identified. These gene pairs are all fragment duplications, and their Ka/Ks values vary from 0.0903 to 0.2191 (Table 1), which are all less than 1, indicating that they are subject to purification selection during the evolution process.

Table 1. Characteristics of LIM genes in alfalfa.

| Gene accession No | Gene | Size (aa) | Molecular weight (D) | Isoelectric point | GRAVY | Subcellular Localization |
|---|---|---|---|---|---|---|
| MS.gene023841.t1 | MsLIM01 | 212 | 23725.85 | 6.20 | -0.602 | cytoplasm |
| MS.gene52210.t1 | MsLIM02 | 212 | 23725.85 | 6.20 | -0.602 | cytoplasm |
| MS.gene27694.t1 | MsLIM03 | 212 | 23725.85 | 6.20 | -0.602 | cytoplasm |
| MS.gene069242.t1 | MsLIM04 | 205 | 22832.79 | 6.78 | -0.449 | cytoplasm |
| MS.gene032012.t1 | MsLIM05 | 191 | 20930.94 | 9.16 | -0.454 | cytoplasm |
| MS.gene008092.t1 | MsLIM06 | 191 | 20930.94 | 9.16 | -0.454 | cytoplasm |
| MS.gene69263.t1 | MsLIM07 | 191 | 20930.94 | 9.16 | -0.454 | cytoplasm |
| MS.gene030430.t1 | MsLIM08 | 211 | 23359.35 | 8.19 | -0.594 | cytoplasm |
| MS.gene048774.t1 | MsLIM09 | 211 | 23354.33 | 8.20 | -0.593 | cytoplasm |
| MS.gene72173.t1 | MsLIM10 | 211 | 23354.33 | 8.20 | -0.593 | cytoplasm |
| MS.gene001795.t1 | MsLIM11 | 160 | 17793.87 | 6.23 | -0.551 | cytoplasm |
| MS.gene056747.t1 | MsLIM12 | 154 | 17137.16 | 6.23 | -0.556 | cytoplasm |
| MS.gene019580.t1 | MsLIM13 | 154 | 17137.16 | 6.23 | -0.556 | cytoplasm |
| MS.gene60342.t1 | MsLIM14 | 181 | 20951.94 | 8.99 | -0.562 | cytoplasm |
| MS.gene033219.t1 | MsLIM15 | 181 | 20920.93 | 9.07 | -0.548 | cytoplasm |
| MS.gene43288.t1 | MsLIM16 | 181 | 20920.93 | 9.07 | -0.548 | cytoplasm |
| MS.gene56949.t1 | MsLIM17 | 181 | 20920.93 | 9.07 | -0.551 | microbody (peroxisome) |
| MS.gene051687.t1 | MsLIM18 | 195 | 21682.81 | 9.06 | -0.609 | mitochondrial matrix space |
| MS.gene66468.t1 | MsLIM19 | 195 | 21682.81 | 9.06 | -0.609 | mitochondrial matrix space |
| MS.gene24483.t1 | MsLIM20 | 195 | 21682.81 | 9.06 | -0.609 | mitochondrial matrix space |
| MS.gene00511.t1 | MsLIM21 | 195 | 21682.81 | 9.06 | -0.609 | mitochondrial matrix space |

## 3.4 Gene structure and basic motif analysis

To determine the distribution of exons and introns of MsLIM genes, this study used the online tool GSDS2.0 to analyze the structure of these genes. The 21 genes of alfalfa are divided into four subfamilies. The number of introns in all members of the αLIM, βLIM, and γLIM subfamily is 4, and the introns of MsLIM11, MsLIM12, and MsLIM13 in the δLIM subfamily are 3 (Fig 2). There are also 4 introns for all 7 members. Although the introns of the LIM gene are different, the members with the highest homology have similar gene structures, with the same number of exons and intron lengths. The motif owned by the gene family or shared by most members is likely to be an indispensable part of the family to perform important functions or structure. For example, it may be the binding site of some sequence-specific proteins (transcription factors) or involve important RNA initiation, termination, shearing, etc. of biological processes. Identifying the common motif of a gene family can understand the characteristics of the gene family so that these characteristics can be used to discover new members of the gene family. It can be seen from Fig 2 that although the conserved motifs of the 21 alfalfa LIM family genes are different in composition, they all contain motif 1 and motif 3 in the same order, with motif 3 first and motif 1 behind, so there is no duplication. MsLIM11, MsLIM12, MsLIM13 contain 5 motifs, MsLIM08, MsLIM09, MsLIM10, MsLIM18, MsLIM19, MsLIM20, and MsLIM21 contain 7 motifs, and the other 11 MsLIM genes all contain 6 motifs. In the

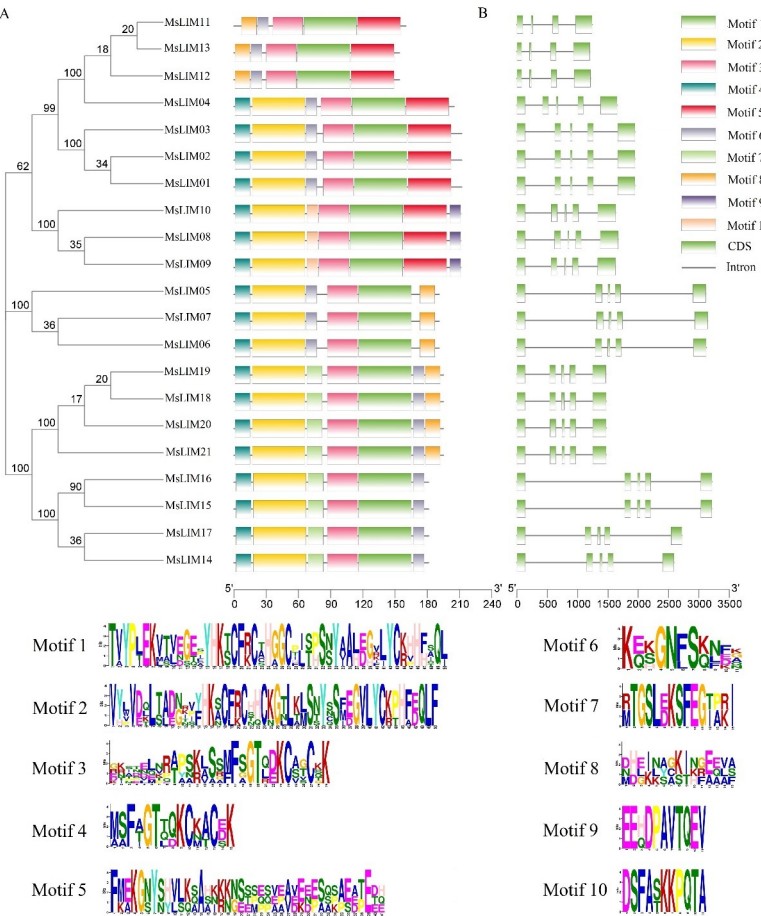

**Fig 2. Structural analysis of MsLIM genes in alfalfa.** (A) The distribution of motif in LIM proteins. (B) The exon-intron structure of LIM gene. (C) The amino acid composition of each motif (S4 Table).

alfalfa LIM gene family, 18 genes are containing motif2, motif4 and motif6. Only MsLIM08, MsLIM09, and MsLIM10 contain motif9. The analysis results indicate that the MsLIM gene family should all contain motif1 and motif3. MsLIM08, MsLIM09, and MsLIM10 are genes with specific functions in the MsLIM gene family. This prediction helps to discover new members of the MsLIM gene family.

### 3.5 Analysis of promoter cis-acting elements

Promoter cis-acting elements are important binding regions for transcription initiation of transcription initiation factors and play an important role in regulating gene expression. To further analyze the possible biological functions of LIM, the 2.0 kb sequence upstream of the alfalfa LIM gene promoter (data in S4 Table) was used to predict cis-acting regulatory elements through the online website Plant CARE. It is predicted that there are many cis-regulatory elements related to transcription, cell cycle, development, hormone, and stress response in the promoter region of the alfalfa LIM gene, and some elements are specific to root specificity, leaf morphology, seed specificity, and meristem specificity (Fig 3). Sex and endosperm specificity are closely related. In addition, we also found many elements related to the hormone signaling pathway, such as methyl jasmonate (MeJA), abscisic acid (ABA), salicylic acid (SA), gibberellin (GA), and auxin (IAA). Among them, 21 LIMs have methyl jasmonate response elements (TGACG-motif and CGTCA-motif), and 37 LIMs have abscisic acid response elements ABRE, which indicates that most LIMs can participate in JA and ABA-mediated signaling pathways. The prediction also found that some elements can interact with various abiotic stresses (drought, salt, heat, cold, light, etc.). It needs to be pointed out that all LIMs contain light response-related components.

### 3.6 Protein interaction network diagram and three-dimensional structure prediction analysis

The use of protein network interactions to connect unknown functional proteins into protein interaction networks will help to further understand the protein biological functions enriched

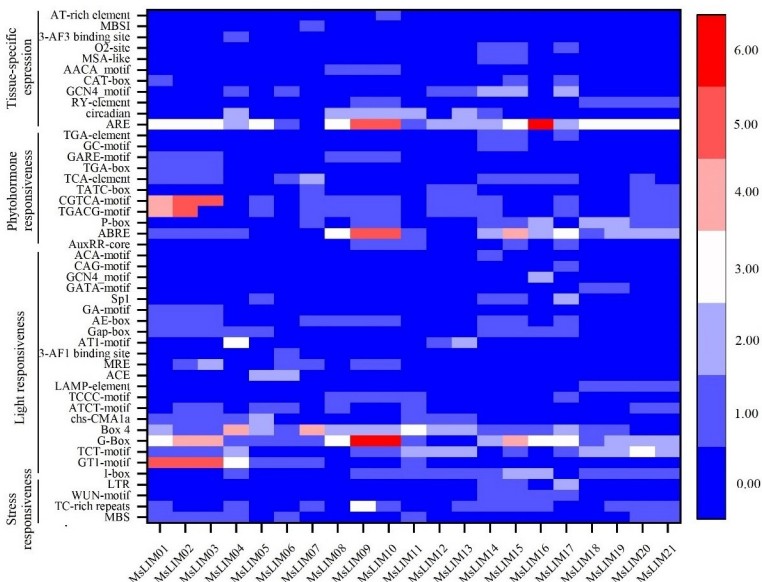

**Fig 3. Promoter analysis of alfalfa LIM genes.**

by protein network interactions and the dynamic network of regulation between various biological molecules in the cell. Therefore, this study used the model plant Arabidopsis as a background to predict the physicochemical properties of LIM protein and the potential interaction proteins related to its function (Fig 4). The expected edge number of our interaction network graph is 10, the average local clustering coefficient is 0.803, and the protein-protein interaction enrichment p-value is <0.00769, so we think the result is reasonable. We identified 3 LIM functioners and 10 potential interacting proteins directly related to the MsLIM protein. They are WLIM1, WLIM2a, PLIM2b, and AT1G01770, AT5G19230, AT5G19250, AT5G19240, DAR7, AT5G52950, AT5G06770, AT4G36860, AT3G06035, and EXPA13. Both WLIM1 and WLIM2a are GATA-type zinc finger transcription factor family proteins. WLIM1 contains 2 LIM domains and WLIM2a contains 1 LIM domain. Both of them can bind, stabilize and bind actin filaments, which indicate that it is involved in the construction and dynamics of actin cytoskeleton structure [9]. Its actin regulating activity is not regulated by pH and $Ca^{2+}$. PLIM2b contains two LIM domains and is also a GATA-type zinc finger transcription factor family protein, which can bind to actin filaments and promote cross-linking into thick bundles, and has an actin stabilizing effect. It inhibits actin regulating activity at pH>6.8 but does not depend on $Ca^{2+}$. From this, we can infer that the functions of 21 LIMS transcription factors of alfalfa are similar to the functions of the above three Arabidopsis transcription factors. Among them, MsLIM01, MsLIM02, MsLIM03, MsLIM04, MsLIM11, MsLIM12, and MsLIM13 have similar functions to PLIM2b, MsLIM05, MsLIM06, MsLIM07, MsLIM08, MsLIM09, and MsLIM10 have similar functions to WLIM2a, MsLIM14, MsLIM15, MsLIM20, MsLIM16, MsLIM17, MsLIM19, and MsLIM21 has similar functions to WLIM1. In this study, the three-dimensional structural homology modeling of the amino acid sequences of 21 MsLIM gene family members of alfalfa was conducted, and the online software Swiss-Model analysis showed that the tertiary structure similarity of the amino acid sequences of 21 members was high. However, the tertiary structure is not exactly the same, which may be related to the length of α-helix, β-sheet, and the length of random coils. These similarities or differences may be the reason for their similar or different functions (Fig 5).

### 3.7 q-PCR analysis

To further determine the expression pattern of the alfalfa MsLIM gene under abiotic stress (low temperature and salt) treatment, q-PCR was used to quantitatively detect the MsLIM gene expression under low temperature and salt stress (S5 Table). In general, the MsLIM genes

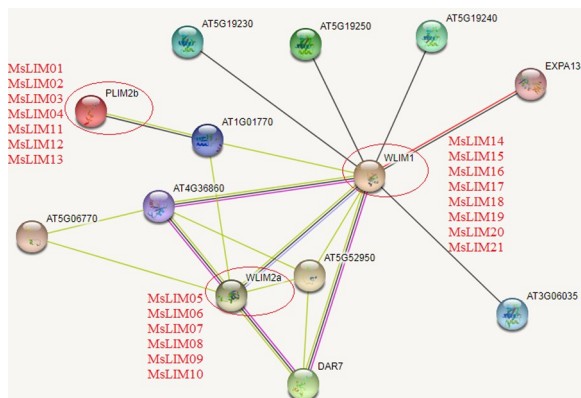

**Fig 4. Protein-protein interactions network structure in *Medicago sativa* L.** The color scales represent the relative signal intensity scores.

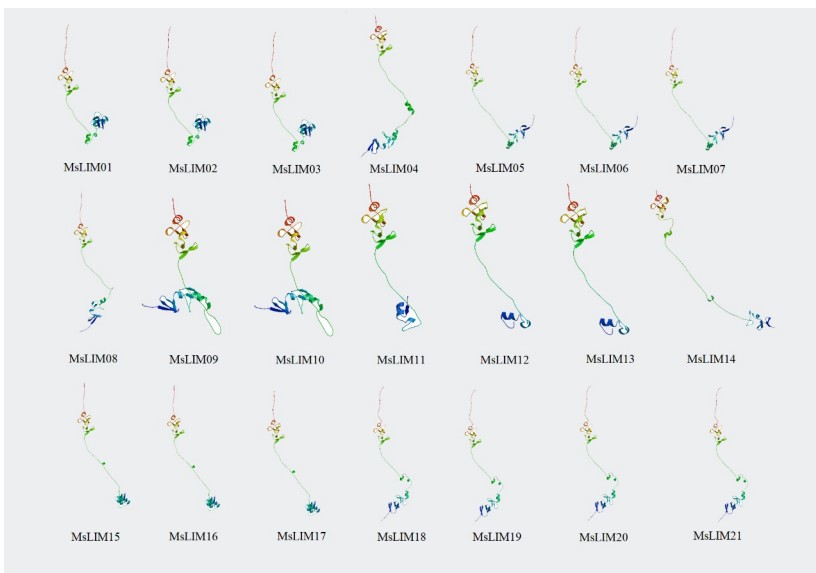

**Fig 5. Tertiary structure of predicted LIM proteins in alfalfa.**

of the four alfalfa families showed different expression patterns, and the expression of these 21 genes could be induced by low temperature and salt treatment. Compared with the control (0h), under low-temperature stress (Fig 6A), the expression levels of MsLIM01, MsLIM02, MsLIM04, and MsLIM09 were significantly up-regulated at two-time points (9h and 12h). However, MsLIM03, MsLIM08, MsLIM10, MsLIM11, MsLIM12, MsLIM13, MsLIM15, MsLIM16, MsLIM17, MsLIM18, MsLIM19, MsLIM20, and MsLIM21 showed the highest expression levels at 3h induced by low temperature, indicating that the stress response of these genes was strong. The expression of MsLIM05, MsLIM06, and MsLIM07 reached the maximum after 6 hours of low-temperature induction. The expression of MsLIM14 at each time point of low-temperature stress was lower than that of the control (0h), indicating that MsLIM14 had a clear response to low-temperature stress. Under salt stress (Fig 6B), MsLIM01-06, MsLIM10, MsLIM12, MsLIM15, and MsLIM18-21 all reached their highest expression levels at 9h. Among them, the expression patterns of MsLIM02, MsLIM04, MsLIM05, MsLIM10, MsLIM12 and MsLIM20 in alfalfa are similar, and the expression level gradually increases from 0-9h to the highest point at 9h, and the expression level decreases in the subsequent experimental period. The expression of MsLIM08, MsLIM09, MsLIM13, MsLIM14, and MsLIM16 peaked at 2h, indicating that these genes had a strong response to salt stress. MsLIM11 has a clear response to salt stress. In addition, the expression level of MsLIM07 was lower than the control (0h) at 3h and 6h, but the expression level increased within 6-12h and reached the maximum at 12h.

## 4 Discussion

The LIM gene family has been studied in many plants, mainly focusing on structure, evolution, growth, and development, etc [33, 34]. However, in alfalfa, there is still a lack of comprehensive and systematic research on the LIM gene family. In 2020, the alfalfa genome is completed [35], which makes it possible to identify all LIM genes of alfalfa and analyze their functions.

In this study, 21 LIM genes were identified from the alfalfa genome. The number of LIM genes is significantly more than that of Arabidopsis (6), rice (6), and poplar (12), which may

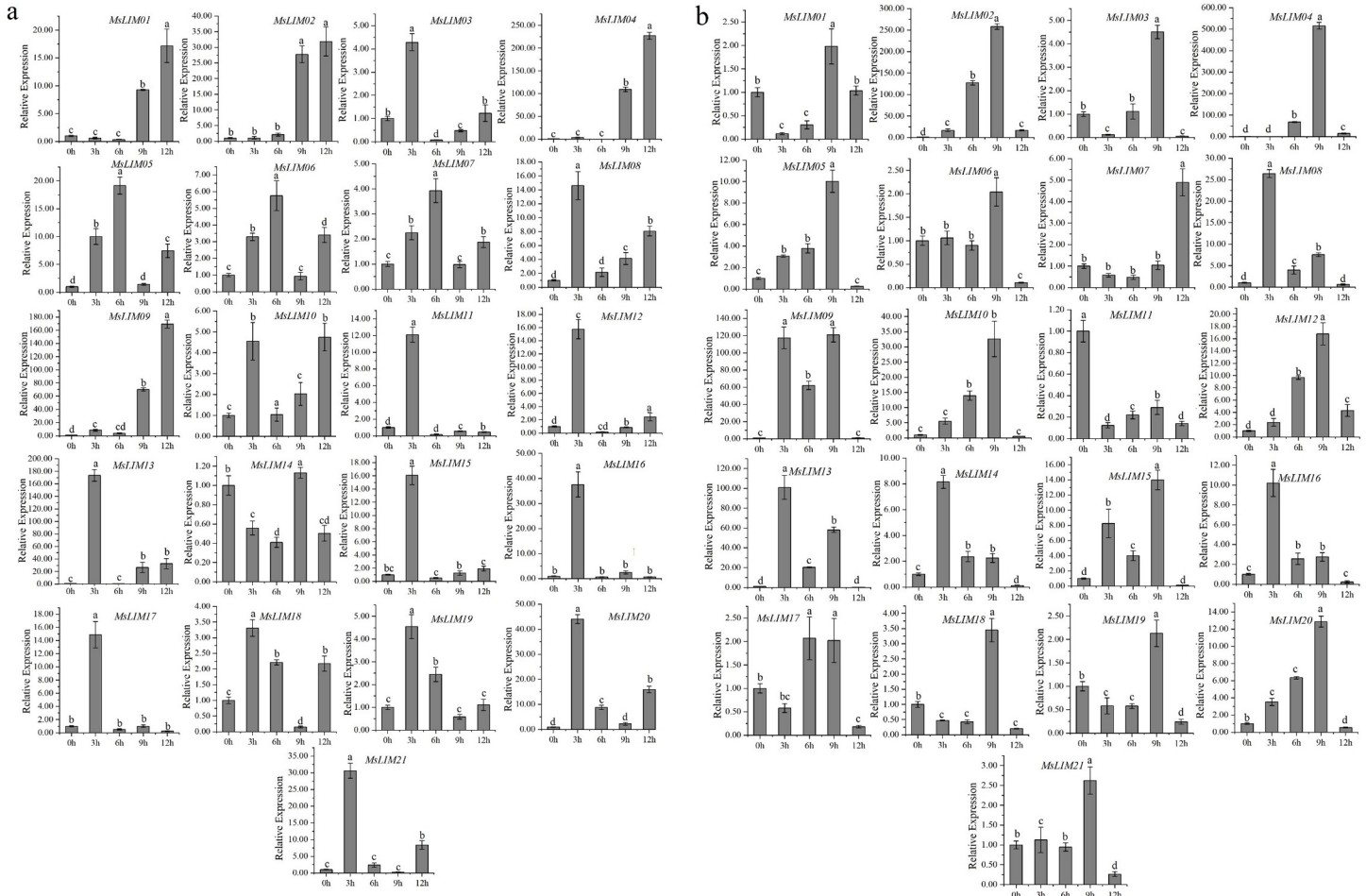

**Fig 6. qPCR expression analysis of LIM genes in *Medicago sativa* L.** q-PCR expression analysis of 21alfalfa LIM genes in response to a) cold and b) NaCl treatment at 0, 3, 6, 9, 12 h of treatment. The vertical bars represent the standard error of the means of three independent replicates. Different letters indicate mean values at different sampling points after Tukey's pair-wise comparison test.

be related to the size of the genome. And multiplication has a certain relationship. The 21 alfalfa LIM gene family members are unevenly distributed on 32 chromosomes (Fig 7), and there is homology among different genes, indicating that the LIM gene duplication or recombination event occurred during the evolution of alfalfa. To understand the gene duplication events of the MsLIM gene in alfalfa, we analyzed the tandem and fragment duplication of the MsLIM gene and found that there are 8 pairs of fragment duplication genes in alfalfa (Table 2), and there is no tandem duplication. MsLIM causes its structure to change during the process of replication, which has a strong influence on the function of genes, indicating that fragment duplication plays an important role in the diversity of MsLIM protein [36].

The phylogenetic tree results show that the closer the clustering relationship, the greater the possibility of having similar functions [37]. In the phylogenetic tree constructed in this study, LIM proteins can be divided into four subfamilies, and the αLIM, βLIM, γLIM and δLIM subfamilies have 4, 4, 3, and 10 genes respectively (Fig 1). In poplar, the LIM gene family can be divided into four sub-families, while in tomato, it is divided into two sub-families. It can be seen that the number of sub-families is different in different species. Through the subcellular

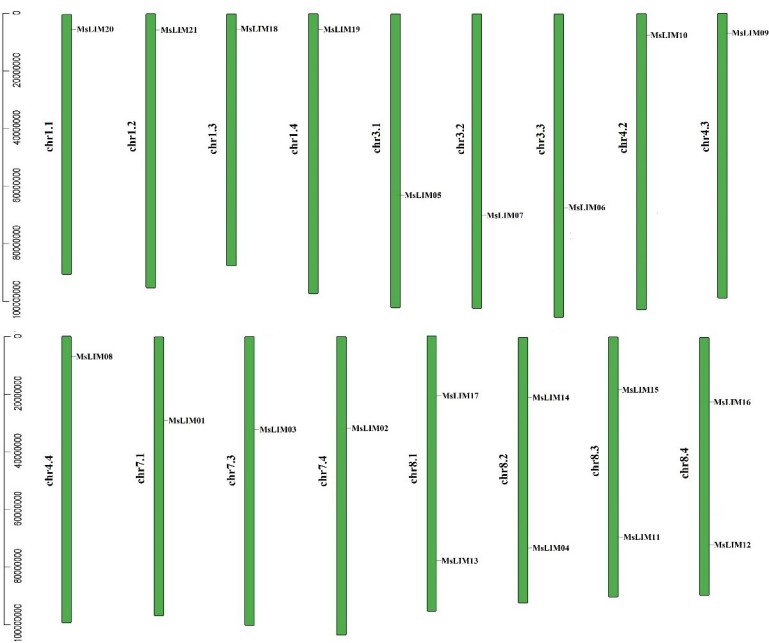

**Fig 7. Chromosome localization of alfalfa LIM family.**

localization analysis of alfalfa LIM protein, it was found that MsLIM01-MsLIM16 is located in the cytoplasm, MsLIM17 is located in the peroxisome, and MsLIM18-MsLIM21 is located in the mitochondrial matrix (Table 1). Studies have shown that the LIM protein located in the cytoplasm is mainly involved in the formation of the cytoskeleton as a regulator of actin [17]. Studies have also shown that the location of LIM proteins in different sub-cells may mean that their functions are also quite different. This conclusion has not yet been confirmed [14]. Gene classification, phylogeny, and subcellular localization analysis help to more accurately and conveniently study the functions of similar gene families. Gene structure analysis showed that most alfalfa MsLIM genes have a similar gene structure, generally having 4 introns and 4–5 exons (Fig 2). This result is similar to the results of previous studies [38], indicating that the family genes are evolutionarily conservative. The structure of individual genes in the alfalfa LIM family is different, and different exon-intron structures also contribute to the diversification of its gene functions [39]. MsLIM protein motifs are highly conserved in alfalfa. These conserved motifs determine the relative conservation of the MsLIM gene function. The difference in gene structure will inevitably lead to the diversity of protein and the difference of function, which is also

**Table 2. Ka/Ks analysis of the MsLIM gene pairs duplication.**

| Duplicated gene1 | Duplicated gene2 | Ka | Ks | Ka/Ks | Purifying selection | Duplicated type |
|---|---|---|---|---|---|---|
| MsLIM17 | MSLIM14 | 0.0094 | 0.0539 | 0.1746 | Yes | Segmental |
| MsLIM17 | MSLIM15 | 0.0094 | 0.0539 | 0.1746 | Yes | Segmental |
| MsLIM17 | MSLIM16 | 0.0094 | 0.0539 | 0.1746 | Yes | Segmental |
| MsLIM10 | MSLIM08 | 0.0040 | 0.0448 | 0.0903 | Yes | Segmental |
| MsLIM09 | MSLIM08 | 0.0040 | 0.0448 | 0.0903 | Yes | Segmental |
| MsLIM03 | MSLIM04 | 0.1419 | 0.6688 | 0.2122 | Yes | Segmental |
| MsLIM02 | MSLIM04 | 0.1421 | 0.6485 | 0.2191 | Yes | Segmental |
| MsLIM01 | MSLIM04 | 0.1419 | 0.6688 | 0.2122 | Yes | Segmental |

the result of genetic evolution. In particular, some genes are missing some motifs. This motif difference may be one of the reasons for the functional diversity of MsLIM genes [40].

Analysis of promoter cis-acting elements showed that MSLIM may be involved in a variety of important biological processes, such as transcription, cell cycle, development, hormones, and biotic/abiotic stress (Fig 3). The MsLIM gene of alfalfa is closely related to hormone pathways and stress response. The main hormones are MEJA, ABA, GA, SA, and IAA. Common stresses include injury, drought, and low-temperature. Tobacco LIM participates in the regulation of cell proliferation and cell cycle processes through gene expression [20]. In addition to pollen-specific expression, the sunflower (*Helianthus annuus* L.) LIM gene also has specific expression in other organs [14]. Research on tomato and olive rape has also proved that LIM can perform multiple functions in tomato [23] and olive rape [41]. In addition, the LIM gene also plays an important role in mediating lignin metabolism in poplar (*Populus alba* L.) [42] and pear (*Pyrus pyrifolia* L.) [24]. It can be seen that LIM is a multifunctional gene family, which is essential for plant growth and development and stress defense response.

The q-PCR quantitative data show under low temperature and salt stress (Fig 6). MsLIM gene can participate in the response process of alfalfa to different stresses. When alfalfa is exposed to low temperature and salt stress, a large number of the MsLIM genes have a strong stress response after 3h of stress. After that, these MsLIM genes are all down-regulated, and some MsLIM genes only respond to one stress. MsLIM14 and MsLIM11 have an obvious response to low-temperature and salt stress. Most of the homologous genes showed similar expression patterns under the above two stresses, indicating that these genes may have similar physiological functions. Some alfalfa MsLIMs exhibit different expression patterns under stress, which may be because they participate in different defense mechanisms against the stress, and more experiments are needed to determine.

Transcription factors play an important role in the process of plant growth and development. By regulating one transcription factor, it is possible to regulate multiple genes with related functions, to achieve the purpose of improving plant traits. In this paper, by analyzing the alfalfa LIM transcription factor and studying its expression under two stresses i.e., low-temperature stress and salt stress, we have a preliminary understanding of the response of this gene in alfalfa after adversity stress, which lays a foundation for its functional verification and will be useful for future cultivation of alfalfa. New varieties of alfalfa and the development of the alfalfa industry are of great significance.

## Supporting information

**S1 Table. The primer designed for qRT-PCR.**
(XLSX)

**S2 Table. The MsLIM protein sequences information.**
(XLSX)

**S3 Table. The LIM protein sequence information in this study.**
(XLSX)

**S4 Table. Consiconserved motifs of MSLIM.**
(XLSX)

**S5 Table. Relative expression of 21 genes by qRT-PCR.**
(XLSX)

**S1 File. Sequence coding for aminoacids in protein of alfalfa.**
(ZIP)

**S2 File. Protein sequence file of alfalfa.**
(ZIP)

**S3 File. Alfalfa genome annotation file.**
(ZIP)

## Author Contributions

**Conceptualization:** Xuelu Liu.

**Formal analysis:** Xianfeng Yi.

**Funding acquisition:** Yingbo Yang.

**Investigation:** Yingbo Yang.

**Resources:** Xiaolin Zhu.

**Supervision:** Xuelu Liu.

**Writing – original draft:** Lili Nian.

**Writing – review & editing:** Fasih Ullah Haider.

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
