## [Decision Letter · Decision Letter 0]

4 Mar 2021

PONE-D-21-00157

Genome-wide identification, phylogenetic and expression analysis under abiotic stress conditions of LIM gene family in Medicago sativa L

PLOS ONE

Dear Dr. nian ,

Thank you for submitting your manuscript to PLOS ONE. After careful consideration, we feel that it has merit but does not fully meet PLOS ONE’s publication criteria as it currently stands. Therefore, we invite you to submit a revised version of the manuscript that addresses the points raised during the review process.

We look forward to receiving your revised manuscript.

Kind regards,

Farrukh Azeem

Academic Editor

PLOS ONE

Journal Requirements:

3. Thank you for stating the following financial disclosure: 'YES'

4. Thank you for stating the following in your Competing Interests section: 'NO'

a. Please complete your Competing Interests statement to state any Competing Interests. If you have no competing interests, please state "The authors have declared that no competing interests exist.", as detailed online in our guide for authors at http://journals.plos.org/plosone/s/submit-now

5. Please ensure that you refer to Figures 2 and 7 in your text as, if accepted, production will need this reference to link the reader to each figure.

6. We note you have included a table to which you do not refer in the text of your manuscript. Please ensure that you refer to Table 2 in your text; if accepted, production will need this reference to link the reader to the Table.

7. Please include captions for your Supporting Information files at the end of your manuscript, and update any in-text citations to match accordingly. Please see our Supporting Information guidelines for more information: http://journals.plos.org/plosone/s/supporting-information

Reviewers' comments:

Reviewer's Responses to Questions

**Comments to the Author**

1. Is the manuscript technically sound, and do the data support the conclusions?

Reviewer #1: Yes

Reviewer #2: Partly

2. Has the statistical analysis been performed appropriately and rigorously? 

Reviewer #1: Yes

Reviewer #2: No

3. Have the authors made all data underlying the findings in their manuscript fully available?

Reviewer #1: Yes

Reviewer #2: Yes

4. Is the manuscript presented in an intelligible fashion and written in standard English?

Reviewer #1: Yes

Reviewer #2: No

5. Review Comments to the Author

Reviewer #1: The authors reported the identification of 21 LIM genes in alfalfa (Medicago sativa L). They examined the genome distribution, protein physicochemical properties, evolutionary relationship, conserved motifs, and responses to abiotic stresses of the MSLIM family by using extensive bioinformatic tools. These analyses generally worked well. However, I have some points that need to be addressed. They need to work on these aspects before this manuscript is accepted.

1)The members of the family are classified according to the classification standards of Arabidopsis and other model plants, Please write down the names of other model plants.

2)The authors used 10 species to study the evolutionary relationship. LIM family members of 9 species except for Arabidopsis, please provide references or websites.

3) It is suggested that subcellular localization should be predicted using multiple websites.

4)The authors did not state the sampled tissue. Furthermore, it is suggested that the expression levels of 21 LIM genes be detected in different tissues.

5) Some formats and grammatical errors need to be modified.

Reviewer #2: Title: Italicize Medicago sativa

Abstract

use MsLIM01-MsLIM21.

“Using bioinformatics methods,”- bad grammar

Write: “physicochemical properties of protein”

Why only salt and low temperature stresses were explored? Include moisture stress, heat and phytohormone treatments in the experiment.

Introduction

First para of introduction: Italicize all scientific names

Third para: include reference in favour of the statement.

Materials and methods

Arabidopsist haliana L??

Italicize all scientific names

“Use Clustal Omega (https://www.ebi.ac.uk/Tools/msa/clustalo/) tool to perform multiple sequence alignment of LIM amino acid sequence.” –example of bad grammar.

Section 1.7: Why relative humidity was that low (40%)? Is there any reference in support? What was the light intensity?

What sample was collected? What was control and what was mock?

How many biological replicates?

Section 1.8 what was the source of independent RNA sample? How many technical replicates?

Discuss statistical procedure in methodology.

Results

“Using the Arabidopsis LIM protein sequence and the alfalfa genome sequence to perform BLAST

comparison to extract the CDS and protein sequence of the alfalfa LIM gene, after de-redundancy

and SMART identification, 21 alfalfa LIM genes were finally identified and named as MSLIM01-

MSLIM21.”- bad grammar, data not cited in favour of the statement.

Expression results: No data was tagged.

Discussion

4 Discussion: capitalize D

Data were not tagged properly

“Conclusion - we have a preliminary understanding of the response of this gene in alfalfa after adversity stress, which lays a foundation for its functional verification and will be useful for future cultivation of alfalfa. New varieties of alfalfa and the development of alfalfa industry in northern my country are of great significance.” –rewrite these sentences.

Tables and Figures

What is SRS in Table 1.

Figure 1: Italicize all scientific names. Soybeans or soybean?

Figure 2. Chromosomal location of LIM family genes. Write MSLIM as MsLIM.

Figure 3. Write MSLIM as MsLIM.

Quality of Figure 4 is very poor. Write MSLIM as MsLIM.

Figure 5: what are ‘these orthologs’?

Figure 7: Italicize Medicago sativa. Some genes were highly expressed under low temperature but some others were highly expressed under NaCl treatment. Prepare separate figures for two different treatments. NaCl, not Nacl.

Professional editing is required.

6. PLOS authors have the option to publish the peer review history of their article (what does this mean?). If published, this will include your full peer review and any attached files.

Reviewer #1: **Yes: **Lifeng Liu

Reviewer #2: No

---

## [Author Response · Author response to Decision Letter 0]

18 Apr 2021

Farrukh Azeem,

Co-Editor in Chief, PLOS One!

Many thanks to you and your precious time on our manuscript (PONE-D-21-00157). The comments and suggestions were helpful in improving the quality of the manuscript. We have revised the manuscript according to the comments and suggestions by you, and are resubmitting for possible publication in PLOS One. We hope the revised manuscript has met the requirements for publication in PLOS One. However, we are open to revising the manuscript further if needed. 

Kind regards,

Lili Nian

Authors

Response to the comments 

Reviewer #1: 

1. The members of the family are classified according to the classification standards of Arabidopsis and other model plants, Please write down the names of other model plants.

Response: Thanks for your valuable suggestion. The authors classify the members of the family according to the classification standards of Arabidopsis tobacco, and rice. Please check lines (104). 

2. The authors used 10 species to study the evolutionary relationship. LIM family members of 9 species except for Arabidopsis, please provide references or websites.

Response: Thanks for your valuable suggestion. The LIM amino acid sequence of Arabidopsis is from TAIR (http://www.arabiodpsis.org). The LIM amino acid sequences of tobacco, barrel medic, soybean, and tomato are from NCBI (https://www.ncbi.nlm.nih.gov/). The LIM amino acid sequences of quinoa, corn, rice, and wheat are from Phytozome (https://phytozome.jgi.doe.gov/pz/portal.html). Please check lines (196-200).

3. It is suggested that subcellular localization should be predicted using multiple websites.

Response: Thanks for your valuable suggestion. We accept the reviewers’ opinions. In addition to using Psort-Prediction (http://psort1.hgc.jp/form.html) to predict subcellular localization, we also use Cell-PLoc (http://www.csbio.sjtu.edu.cn/bioinf/Cell-PLoc-2/) predicts the subcellular localization of LIM protein. Please check lines (93-94). 

4. The authors did not state the sampled tissue. Furthermore, it is suggested that the expression levels of 21 LIM genes be detected in different tissues.

Response: Thank you for pointing this out. We take the mature leaves of alfalfa as samples. As the reviewer said, we should detect the expression levels of 21 LIM genes in different tissues of alfalfa. However, on the one hand, our experimental funds are limited and it is difficult to support us to detect the expression levels of these 21 LIM genes in other tissues. On the other hand, the author only wanted to focus on the expression patterns of these 21 LIM genes in alfalfa leaves in this study. Thank you again for your comments, we are willing to adopt them in subsequent experiments.

5. Some formats and grammatical errors need to be modified.

Response: We are very sorry for our incorrect formats and grammatical. We have polished the language of the full text. At the same time, the wrong format was modified. Further, the manuscript has been critically edited by Prof. Dr. Muhammad Farooq, from Sultan Qaboos University, Oman, who has relevant work experience and publications.

Reviewer #2:

1. Title: Italicize Medicago sativa

Response: Thanks for suggestion; we have made correction according to the Reviewer’s comments in whole manuscript.

2. Abstract

use MsLIM01-MsLIM21.

Response: We have modified all MSLIM that appear in the full text to MsLIM.

3. “Using bioinformatics methods,”- bad grammar

Response: Sorry for inconvenience. Considering the Reviewer’s suggestion, we have modified “Using bioinformatics methods,” to “Comprehensive analysis of the chromosome location, protein physicochemical properties, evolutionary relationship, conserved motifs, and responses to abiotic stresses of the LIM gene family in alfalfa using bioinformatics methods”. Please check lines (15-17).

4. Write: “physicochemical properties of protein”

Response: We have made correction according to the Reviewer’s comments. Modify “protein physicochemical properties” to “physicochemical properties of protein”. Please check lines (16).

5. Why only salt and low temperature stresses were explored? Include moisture stress, heat and phytohormone treatments in the experiment.

Response: We only choose low-temperature and salt treatment because we refer to previous studies on tomatoes and rape. Studies have found that the LIM family responds significantly to the regulation of low-temperature and salt stress. These two studies will appear in the attachment to my revised manuscript.

Khatun K, Robin AHK, Park JI, Ahmed NU, Kim CK, Lim KB, Kim MB, Lee DJ, Nou IS, Chung MY. Genome-wide identification, characterization and expression profiling of LIM family genes in Solanum lycopersicum L. Plant Physiol. Biochem. 2016;108:177-190. doi: 10.1016/j.plaphy.2016.07.006. PMID: 27439220.

Park JI, Ahmed NU, Jung HJ, Arasan SK, Chung MY, Cho YG, Watanabe M, Nou IS. Identification and characterization of LIM gene family in Brassica rapa. BMC Genomics. 2014;15(1):641. doi: 10.1186/1471-2164-15-641. PMID: 25086651.

6. Introduction

First para of introduction: Italicize all scientific names

Response: Thanks for suggestion; all scientific names in the first paragraph of the introduction have been italicized.

7. Third para: include reference in favour of the statement.

Response: Considering the Reviewer’s suggestion, we have added the references in the third paragraph. Please check lines (71). 

8. Materials and methods

Arabidopsis thaliana L??

Response: We are very sorry for our incorrect writing. And we have modified “Arabidopsis Breeding Toolbox” to “Alfalfa Breeder's Toolbox”. Please check lines (81).

9. Italicize all scientific names

Response: Done. Please check lines (96-99).

10. “Use Clustal Omega (https://www.ebi.ac.uk/Tools/msa/clustalo/) tool to perform multiple sequence alignment of LIM amino acid sequence.” –example of bad grammar.

Response: We have re-written this part according to the Reviewer’s suggestion. We have modified “Use Clustal Omega (https://www.ebi.ac.uk/Tools/msa/clustalo/) tool to perform multiple sequence alignment of LIM amino acid sequence.” to “Fast, scalable generation of LIM protein multiple sequence alignments using Clustal Omega(https://www.ebi.ac.uk/Tools/msa/clustalo/)”. Please check lines (99-100).

11. Section 1.7: Why relative humidity was that low (40%)? Is there any reference in support? What was the light intensity?

Response: We are very sorry for our incorrect writing,the relative humidity should be 80%.And we set the light intensity to 4000 lx.

12. What sample was collected? What was control and what was mock?

Response: We take the mature leaves of alfalfa as samples. Control refers to the relative expression of 21 MsLIM genes at 0h, and mock refers to the relative expression of 21 MsLIM genes at each time period under low-temperature and salt treatment.

13. How many biological replicates?

Response: We did three biological replicates.Please check lines (137).

14. Section 1.8 what was the source of independent RNA sample? How many technical replicates?

Response: We picked the mature leaves of alfalfa at different time periods under the two treatments as samples to extract RNA. We have three technical replicates. Please check lines (137,150). 

15. Discuss statistical procedure in methodology.

Response: We have added this part according to the Reviewer’s suggestion. Please check lines (150-155).

16. Results

“Using the Arabidopsis LIM protein sequence and the alfalfa genome sequence to perform BLAST

comparison to extract the CDS and protein sequence of the alfalfa LIM gene, after de-redundancy

and SMART identification, 21 alfalfa LIM genes were finally identified and named as MSLIM01-

MSLIM21.”- bad grammar, data not cited in favour of the statement.

Expression results: No data was tagged.

Response: Thanks for suggestion; we have revised these results for further clarity. Please check lines (158-162). 

17. Discussion: capitalize D

Response: We are sorry for this kind of mistakes to eliminate such kind of problems. This problem has now been modified. Please check lines (332). 

18. Data were not tagged properly

Response: Considering the Reviewer’s suggestion, we have tagged data. 

19. “Conclusion - we have a preliminary understanding of the response of this gene in alfalfa after adversity stress, which lays a foundation for its functional verification and will be useful for future cultivation of alfalfa. New varieties of alfalfa and the development of alfalfa industry in northern my country are of great significance.” –rewrite these sentences.

Response: Done.

20. Tables and Figures

What is SRS in Table 1.

Response: We are sorry for this kind of mistakes to eliminate such kind of problems. We have modified SRS to LIM. Please check lines (169)

21. Figure 1: Italicize all scientific names. Soybeans or soybean?

Response: We have made correction according to the Reviewer’s comments. And modify Soybeans to soybean. Please check lines (192)

22. Figure 2. Chromosomal location of LIM family genes. Write MSLIM as MsLIM.

Response: We have made correction according to the Reviewer’s comments. Please check Figure 2.

23. Figure 3. Write MSLIM as MsLIM.

Response: We have made correction according to the Reviewer’s comments. Please check Figure 3.

24. Quality of Figure 4 is very poor. Write MSLIM as MsLIM.

Response: We have made correction according to the Reviewer’s comments. And have re-draw Figure 4.

25. Figure 5: what are ‘these orthologs’?

Response: We are very sorry for our incorrect writing. And we have re-written the name of Figure 5. Please check line (297-298).

26. Figure 7: Italicize Medicago sativa. Some genes were highly expressed under low temperature but some others were highly expressed under NaCl treatment. Prepare separate figures for two different treatments. NaCl, not Nacl.

Response: Thanks for suggestion. We have separated figures for two different treatments, which will help readers to better understand this article. Further, Medicago sativa has been modified to Medicago sativa, and Nacl has been modified to NaCl. Please check Figure 7.

---

## [Decision Letter · Decision Letter 1]

12 May 2021

Genome-wide identification, phylogenetic and expression analysis under abiotic stress conditions of LIM gene family in Medicago sativa L

PONE-D-21-00157R1

Dear Dr. nian ,

We’re pleased to inform you that your manuscript has been judged scientifically suitable for publication and will be formally accepted for publication once it meets all outstanding technical requirements.

Kind regards,

Farrukh Azeem

Academic Editor

PLOS ONE

Reviewers' comments:

Reviewer's Responses to Questions

**Comments to the Author**

1. If the authors have adequately addressed your comments raised in a previous round of review and you feel that this manuscript is now acceptable for publication, you may indicate that here to bypass the “Comments to the Author” section, enter your conflict of interest statement in the “Confidential to Editor” section, and submit your "Accept" recommendation.

Reviewer #1: All comments have been addressed

Reviewer #2: All comments have been addressed

2. Is the manuscript technically sound, and do the data support the conclusions?

Reviewer #1: Yes

Reviewer #2: Yes

3. Has the statistical analysis been performed appropriately and rigorously? 

Reviewer #1: Yes

Reviewer #2: Yes

4. Have the authors made all data underlying the findings in their manuscript fully available?

Reviewer #1: Yes

Reviewer #2: Yes

5. Is the manuscript presented in an intelligible fashion and written in standard English?

Reviewer #1: Yes

Reviewer #2: Yes

6. Review Comments to the Author

Reviewer #1: (No Response)

Reviewer #2: Italicize botanical names in the reference section and revise references carefully. Carefully edit the manuscript for spacing and other typos.

7. PLOS authors have the option to publish the peer review history of their article (what does this mean?). If published, this will include your full peer review and any attached files.

Reviewer #1: No

Reviewer #2: **Yes: **Arif Hasan Khan Robin

---

## [Editor Report · Acceptance letter]

17 Jun 2021

PONE-D-21-00157R1 

Genome-wide identification, phylogenetic, and expression analysis under abiotic stress conditions of LIM gene family in *Medicago sativa L.*

Dear Dr. Nian:

I'm pleased to inform you that your manuscript has been deemed suitable for publication in PLOS ONE. Congratulations! Your manuscript is now with our production department. 

Kind regards, 

on behalf of

Dr. Farrukh Azeem 

Academic Editor

PLOS ONE